# Exploring the effect of hypertension on retinal microvasculature using deep learning on East Asian population

Guangzheng Dai[1,2], Wei He[1,2,3]*, Ling Xu[2,3], Eric E. Pazo[2,3], Tiezhu Lin[2,3], Shasha Liu[2], Chenguang Zhang[1,2]

1 The Second Clinical College, Dalian Medical University, Dalian, Liaoning, China, 2 Clinical Research Center, He Eye Specialists Hospitals, Shenyang, Liaoning, China, 3 Clinical College, He University, Shenyang, Liaoning, China

* hsyk2017@163.com

**Data Availability Statement:** The data underlying this study have been uploaded to DRYAD and are accessible using the following doi: https://doi.org/10.5061/dryad.280gb5mmh.

## Abstract

Hypertension is the leading risk factor of cardiovascular disease and has profound effects on both the structure and function of the microvasculature. Abnormalities of the retinal vasculature may reflect the degree of microvascular damage due to hypertension, and these changes can be detected with fundus photographs. This study aimed to use deep learning technique that can detect subclinical features appearing below the threshold of a human observer to explore the effect of hypertension on morphological features of retinal microvasculature. We collected 2012 retinal photographs which included 1007 from patients with a diagnosis of hypertension and 1005 from normotensive control. By method of vessel segmentation, we removed interference information other than retinal vasculature and contained only morphological information about blood vessels. Using these segmented images, we trained a small convolutional neural networks (CNN) classification model and used a deep learning technique called Gradient-weighted Class Activation Mapping (Grad-CAM) to generate heat maps for the class "hypertension". Our model achieved an accuracy of 60.94%, a specificity of 51.54%, a precision of 59.27%, and a recall of 70.48%. The AUC was 0.6506. In the heat maps for the class "hypertension", red patchy areas were mainly distributed on or around arterial/venous bifurcations. This indicated that the model has identified these regions as being the most important for predicting hypertension. Our study suggested that the effect of hypertension on retinal microvascular morphology mainly occurred at branching of vessels. The change of the branching pattern of retinal vessels was probably the most significant in response to elevated blood pressure.

## Introduction

Cardiovascular diseases are the world's biggest killers and these diseases have remained the leading causes of death globally in the last 15 years [1]. Hypertension is the leading modifiable risk-factor, which affects 23.2% (estimated 244.5 million) of Chinese adult population aged

**Funding:** The authors received no specific funding for this work.

**Competing interests:** The authors have declared that no competing interests exist.

≥18 years [2]. There is evidence that elevated blood pressure has a substantial impact on the microvasculature in end-organs, such as the brain, heart, kidney, eye and so on [3–7]. In particular, retinal vasculature, measuring 100 to 300 μm in size, has attracted a lot of non-ophthalmological attention. Retinal microvascular abnormalities represent a manifestation of ongoing systemic microvascular damage and can be viewed directly and noninvasively, offering a unique and easily accessible "window" to study the human microcirculation in vivo. Advances in digital retinal photography and computer image analysis have now enabled more objective quantitative assessment of retinal microvascular structure and function, and may offer a potential noninvasive research tool to assess the pathophysiology of hypertension.

Extensive researches on retinal microvascular phenotypes in fundus images have shown that hypertension can lead to abnormal signs on retina [8–28]. These abnormal signs can be broadly divided into four groups: classic hypertensive retinopathy, isolated retinopathy (e.g., retinal hemorrhage, microaneurysm, or cotton wool spot), changes from retinal vascular caliber (e.g., generalized retinal arteriolar narrowing, focal arteriolar narrowing, arteriovenous nicking), and changes from retinal vascular architecture (e.g., retinal tortuosity, fractal dimension, branching angle). These signs probably reflect systemic microvascular damage and may be an early indicator of cardiovascular diseases. In addition, some prospective studies suggest that retinal microvascular abnormal signs are predictive of the subsequent risk of hypertension independently of other known risk factors [29–35]. Although a large number of studies have reflected the association between abnormal retinal microvascular signs and hypertension, some results were inconsistent with three reasons. Firstly, qualitative assessment of hypertensive retinopathy is mainly based on the experiences of the individual and the evaluation results lacks objectivity. Secondly, there is a variety of methods of computer-assisted quantification of retinal vasculature, such as retinal vessel caliber, and thus measurements given for the same fundus image often vary. Last but not least, variations in image brightness, focus, and contrast can significantly affect the measurement of retinal vasculature.

Thus this study was designed to analyze retinal image using convolutional neural networks (CNN), also known as convnets, a type of deep-learning model almost universally used in computer vision applications. One fundamental characteristic of convnets that is composed of multiple processing layers is that it can find interesting features in training data on its own, without any need for manual feature engineering [36]. This is especially useful in problems where the input samples are very high-dimensional, like retinal fundus images. It can detect subclinical and discrete features appearing below the threshold of a human observer and quantify minimal differences in feature expression. Recently, a CNN was trained on fundus images to screen DR [37–42] or age-related macular degeneration [43]. Moreover, a more sophisticated CNN (Google Inceptionv3) has been trained on datasets from the UK Biobank [44] and EyePACS [45] cohorts to detect cardiovascular risk factors from retinal images such as age, gender, hypertension, and smoking status [46]. In this paper, we have constructed an automated segmentation model to delineate retinal vascular structure of fundus photographs and introduced visualization technique to further explore pathophysiological changes of retinal microvasculature in hypertensive patients.

## Materials and methods

### Data collection

The research followed the tenets of the Declaration of Helsinki and was reviewed by the Committee on Medical Ethics of Shenyang He Eye Hospital. We collected 735 patients (1007 eyes) with a diagnosis of hypertension and 684 normotensive control subjects (1005 eyes), who were admitted between May 2017 and December 2018 to Shenyang He Eye Hospital with eye

disease. The case group included 274 males and 461 females, mean age 66.47 ± 9.06 years (from 29 to 92 years), whereas the control group was composed of 326 males and 358 females, mean age 61.27 ± 10.48 years (from 31 to 88 years). Age of hypertension group was higher than control group ($P < 0.001$), and the proportion of women is higher ($P < 0.001$). Individual written informed consent could not be obtained, because all the subjects were admitted to Shenyang He Eye Hospital in 2017–2018 and most of them could not be contacted. In addition, our dataset did not contain any patient identification information. We applied to the Committee on Medical Ethics for exemption from informed consent and obtained permission.

Each patient underwent ophthalmological examination and standard assessments of cardiovascular risk factors. Retinal photographs that were centered on the macula and documented the optic disc, the macula, substantial portions of the temporal vascular arcades were taken with 45˚non-mydriatic digital camera (TRC-NW300, Topcon, Tokyo, Japan) after dilation of the pupils with tropicamide phenylephrine eye drops. Hypertension was defined as systolic blood pressure greater than 140 mm Hg, diastolic blood pressure above 90 mmHg, or use of antihypertensive medication during the previous 2 weeks. Exclusion criteria were: diabetes mellitus that was defined as a fasting blood glucose concentration above 7.0 mmol/L, a nonfasting value of more than 11.1 mmol/L, or a self-reported history of treatment for diabetes; poor dilation or ocular media opacities so that part of or the entire retinal vessel is almost indiscernible (e.g., cataracts with high-severity opacity of lens, vitreous hemorrhage); any other previous or coexisting ocular disease that could affect the retinal vasculature (e.g., glaucoma, central or branch retinal artery occlusion, central or branch retinal vein occlusion).

## Image preprocessing

The JPEG content was decoded to RGB grids of pixels at a size of 2048×1536. For pre-processing, auto-cutting was employed to minimize black borders around the field of view and yields square images, of dimension1496×1496, in order to scale the images without distortion. And then all images were resized to 565×565. To highlight the retinal vasculature, two image enhancement methods, gamma correction and contrast limited adaptive histogram equalization (CLAHE), were conducted to enhance the image contrast, and "enhanced dataset" was obtained (Figs 1 and 2).

CLAHE is a computer image processing technique used to improve the image contrast [47]. Briefly, it computes several histograms corresponding to distinct sections of the image and limits the amplification by clipping the histograms at a predefined value, and uses them to redistribute the lightness values of the image. It is therefore suitable for improving the local contrast of an image and bringing out more detail.

Gamma correction is a nonlinear mapping of the gray value of the input image and can be expressed by the equation [48].

$$I_\gamma = pI^\gamma$$

$I$ is the input intensity, $I_\gamma$ is the output intensity, and $p$ is a normalization factor which is determined by the value of $\gamma$. In this study, the setting of $\gamma$ is equal to 1/1.2, and the overall brightness of the image is improved, while the contrast at the low gray level is increased, which is more conducive to the image details at the low gray level.

## Retinal vessel segmentation

The vessel segmentation task was based on an open source hosted at GitHub [49]. Briefly, the automated vessel segmentation model is derived from the U-Net architecture [50] and was

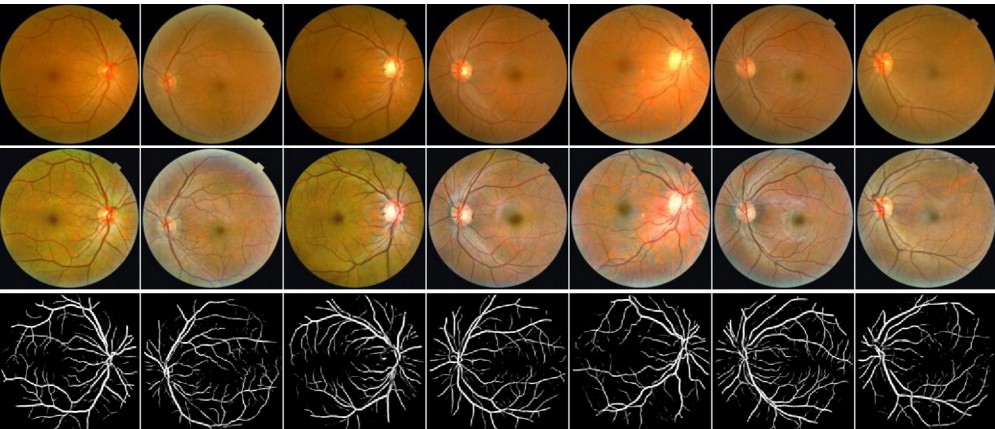

**Fig 1. Examples of fundus photographs from normotensive subjects.** Top row: Raw retinal images. Middle row: enhanced images. Bottom row: segmented blood vessel images.

trained on DRIVE (digital retinal images for vessel extraction) dataset. The DRIVE is a publicly available database, containing a total of 40 colored fundus photographs obtained from a diabetic retinopathy (DR) screening program in the Netherlands [51]. There are 20 images used for training and the rest 20 for testing by default. The performance of this segmentation model is tested on the DRIVE database, and it achieves 0.9790 of area under the ROC curve. On STARE database, it achieves 0.9805 of area under the ROC curve. We used the automated segmentation model to extract retinal vessels of every fundus image of "enhanced dataset", and "segmented dataset" was obtained (Figs 1 and 2).

## Classification model development

A small CNN architecture that was used for training and classification in our project is presented below Fig 3. In short, five convolution layers, five pooling layers and two fully-connected layers composed the main body of CNN. To prevent overfitting, the size of the model, namely learnable parameters in the model, was reduced as much as possible. The number of filters in the convolution layers and the number of units in the densely-connected layers was

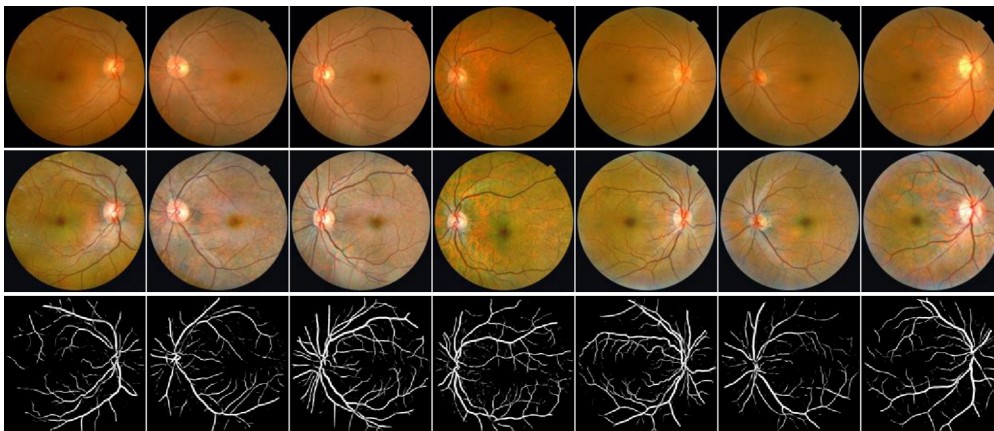

**Fig 2. Examples of fundus photographs from hypertension patients.** Top row: Raw retinal images. Middle row: enhanced images. Bottom row: segmented blood vessel images.

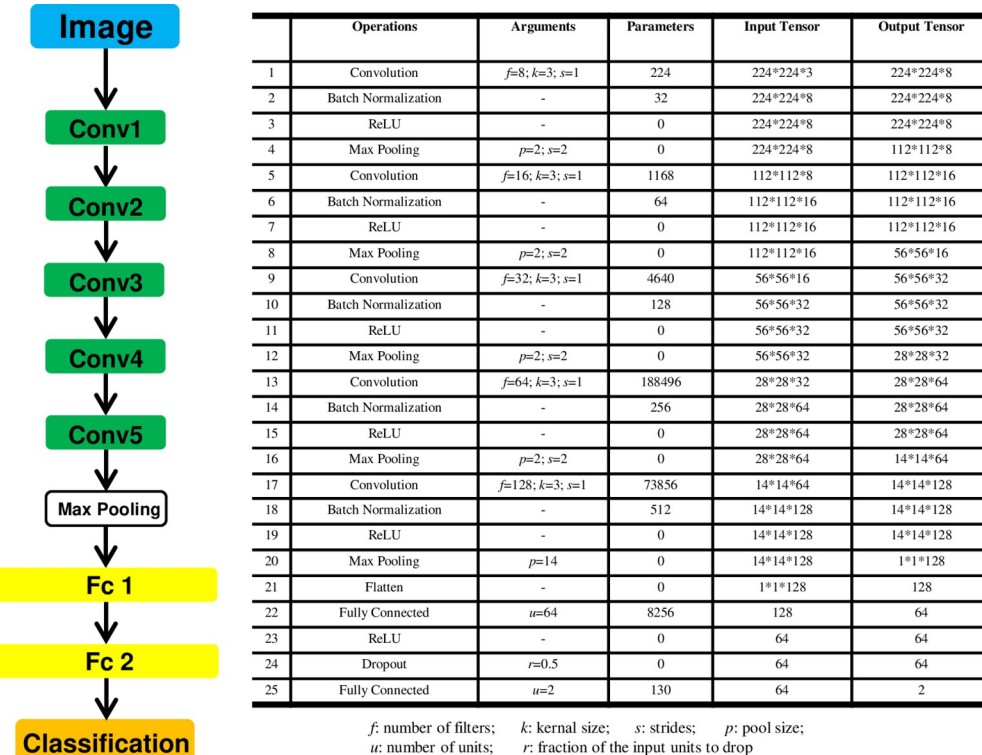

| | Operations | Arguments | Parameters | Input Tensor | Output Tensor |
|---|---|---|---|---|---|
| 1 | Convolution | $f$=8; $k$=3; $s$=1 | 224 | 224*224*3 | 224*224*8 |
| 2 | Batch Normalization | - | 32 | 224*224*8 | 224*224*8 |
| 3 | ReLU | - | 0 | 224*224*8 | 224*224*8 |
| 4 | Max Pooling | $p$=2; $s$=2 | 0 | 224*224*8 | 112*112*8 |
| 5 | Convolution | $f$=16; $k$=3; $s$=1 | 1168 | 112*112*8 | 112*112*16 |
| 6 | Batch Normalization | - | 64 | 112*112*16 | 112*112*16 |
| 7 | ReLU | - | 0 | 112*112*16 | 112*112*16 |
| 8 | Max Pooling | $p$=2; $s$=2 | 0 | 112*112*16 | 56*56*16 |
| 9 | Convolution | $f$=32; $k$=3; $s$=1 | 4640 | 56*56*16 | 56*56*32 |
| 10 | Batch Normalization | - | 128 | 56*56*32 | 56*56*32 |
| 11 | ReLU | - | 0 | 56*56*32 | 56*56*32 |
| 12 | Max Pooling | $p$=2; $s$=2 | 0 | 56*56*32 | 28*28*32 |
| 13 | Convolution | $f$=64; $k$=3; $s$=1 | 188496 | 28*28*32 | 28*28*64 |
| 14 | Batch Normalization | - | 256 | 28*28*64 | 28*28*64 |
| 15 | ReLU | - | 0 | 28*28*64 | 28*28*64 |
| 16 | Max Pooling | $p$=2; $s$=2 | 0 | 28*28*64 | 14*14*64 |
| 17 | Convolution | $f$=128; $k$=3; $s$=1 | 73856 | 14*14*64 | 14*14*128 |
| 18 | Batch Normalization | - | 512 | 14*14*128 | 14*14*128 |
| 19 | ReLU | - | 0 | 14*14*128 | 14*14*128 |
| 20 | Max Pooling | $p$=14 | 0 | 14*14*128 | 1*1*128 |
| 21 | Flatten | - | 0 | 1*1*128 | 128 |
| 22 | Fully Connected | $u$=64 | 8256 | 128 | 64 |
| 23 | ReLU | - | 0 | 64 | 64 |
| 24 | Dropout | $r$=0.5 | 0 | 64 | 64 |
| 25 | Fully Connected | $u$=2 | 130 | 64 | 2 |

$f$: number of filters;    $k$: kernal size;    $s$: strides;    $p$: pool size;
$u$: number of units;    $r$: fraction of the input units to drop

**Fig 3. CNN architecture.** $f$: number of filters; $k$: kernel size; $s$: strides; $p$: pool size; $u$: number of units; $r$: fraction of the input units to drop.

set to a smaller value and global max pooling layer was used after the last convolution layer. Batchnormalization layers were used for accelerating converge, and dropout layers were added to combat overfitting.

For either of the two datasets ("enhanced dataset" and "segmented dataset"), all fundus images with one target label: hypertension-status (Yes/No) were randomly partitioned into five equal sized subsamples. Of the five subsamples, a single subsample was retained as the "test set" which was not used during the training process to test the model, and the remaining four subsamples are used as the "development set" to develop our model. The training process was then repeated five times.

During the training process, the "development set" was divided into a training set (75%) and a validation set (25%) which was a random subset of the "development set" and was not used to train the model parameters, but was used as a small evaluation dataset for tuning the model. The training set was resized to 256×256, and via a number of random transformations, which include adjusting the brightness of images by random factor, flipping images horizontally or vertically, randomly cropping images to 224×224, yielded a number of believable-looking images. Thus, at each iteration, the model never saw the exact same picture twice. This helps expose the model to more aspects of the data and generalize better. The validation set were resized to 224×224, and data augmentation techniques was not applied during the validation process.

All codes employed in our study were executed in the tensorflow2.0 framework with Windows10 + CUDA (Compute Unified Device Architecture) 10.0. Experiments were run in an Intel Core i9-9900K CPU @ 3.60 GHz with 32.0 GB of RAM memory and a NVIDIA GeForce RTX 2080.

## Evaluating the model

Due to the small sample size of the dataset, five-fold cross-validation was adopted. The cross-validation process was repeated five times, and the five results were averaged to produce a single estimation. In order to keep the image size consistent with cropped pictures during the training process, all testing images were resized to 224×224. To evaluate performances of the learned model, we adopted several evaluation parameters, including accuracy, specificity, precision, recall, and the area under the receiver operating characteristic curve (AUC).

$$Accuracy = \frac{TP + TN}{TP + TN + FP + FN}$$

$$Specificity = \frac{TN}{TN + FP}$$

$$Precision = \frac{TP}{TP + FP}$$

$$Recall = \frac{TP}{TP + FN}$$

*TP* represents correctly classified hypertension images and *TN* represents correctly classified non-hypertension images. *FP* represents false positive values, where non-hypertension images are wrongly classified as hypertension and *FN* represents false negative values where hypertension images are wrongly classified as non-hypertension.

## Visualizing heat maps of class activation

To better understand which parts of a given retinal photograph led a CNN to its final classification decision, we used a deep learning technique called Gradient-weighted Class Activation Mapping (Grad-CAM) [52]. This approach uses the gradients of target concept, flowing into the final convolutional layer to produce a coarse localization map highlighting the important regions in the image for predicting the concept. Heat maps for the class "hypertension" or "non-hypertension" were generated from the "segmented dataset" as images from this dataset contained only morphological attributes of retinal blood vessels, such as length, width, tortuosity and branching pattern. After removing interference information other than blood vessels, we can accurately find the effect of hypertension on retinal blood vessels.

# Results and discussion

## Results

Both "enhanced dataset" and "segmented dataset" were used independently, to train and test a CNN model. Although methods, such as reducing the size of the model, using dropout layers, and data augmentation techniques were used, these couldn't entirely eliminate overfitting. For "enhanced dataset", with the decreasing of the training loss (cross-entropy loss between true labels and predict labels), the drop in validation loss is quite small. The three hundred epochs training-stop criterion was chosen as it was observed that the model validation loss started stalling (or increasing) after 300 epochs of training. For "segmented dataset", the model started overfitting, and its performance degrades more slowly once it starts overfitting. The five hundred epochs training-stop criterion was chosen as it started stalling (or increasing) after 500 epochs of training.

**Table 1. The performance of the model on two different dataset.**

| | enhanced dataset | | | | | segmented dataset | | | | |
|---|---|---|---|---|---|---|---|---|---|---|
| **Accuracy** | 53.35% | 56.47% | 57.21% | 59.31% | 57.46% | 58.31% | 60.20% | 63.68% | 59.31% | 63.18% |
| | Average 56.76% | | | | | Average 60.94% | | | | |
| **Specificity** | 51.04% | 56.48% | 76.44% | 80.32% | 54.73% | 52.60% | 48.61% | 54.81% | 49.47% | 52.23% |
| | Average 63.80% | | | | | Average 51.54% | | | | |
| **Precision** | 55.45% | 52.76% | 59.17% | 70.40% | 57.08% | 59.56% | 55.24% | 60.17% | 60.58% | 60.82% |
| | Average 58.97% | | | | | Average 59.27% | | | | |
| **Recall** | 55.45% | 56.45% | 36.60% | 40.93% | 60.20% | 63.51% | 73.66% | 73.20% | 67.91% | 74.13% |
| | Average 49.93% | | | | | Average 70.48% | | | | |
| **AUC** | 0.5572 | 0.6144 | 0.6083 | 0.6532 | 0.6014 | 0.5893 | 0.6634 | 0.6655 | 0.6558 | 0.6789 |
| | Average 0.6069 | | | | | Average 0.6506 | | | | |

For "enhanced dataset", averaged the results of the five cross-validation and our model achieved an accuracy of 56.76%, a specificity of 63.80%, a precision of 58.97%, and a recall of 49.93%. The AUC was 0.6069. For "segmented dataset", our model produced an improved accuracy 60.94% and recall 70.48%, a similar precision 59.27%, but its specificity dropped to 51.54%. Finally, the AUC was 0.6506 (Table 1).

Using the "segmented dataset", the learned model had the highest prediction precision for hypertension at the last time of repeated cross-validation. So we used the model trained from the last cross-validation and corresponding "test set" to produce heat maps for the class "hypertension" and "non-hypertension". There are two hundred and fifty heat maps of class activation, one hundred and thirty-six from correctly classified hypertension images and one hundred and fourteen from correctly classified non-hypertension images, and representative examples are shown in Figs 4 and 5. For heat maps for the class "hypertension", red patchy areas that were strongly activated showed discrete distribution, and most of them were on or around arterial/venous bifurcations. This is how the network can tell the difference of retinal microvascular morphology between hypertension and non-hypertension. But, for heat maps for the class "non-hypertension", red areas showed a continuous distribution along the blood vessels.

## Discussion

In our study, the dataset size was small for deep learning. Although we use some strategies for mitigating overfitting and maximizing generalization, such as reducing the size of the model, using dropout layers, and data augmentation, the model eventually started over-fitting after a certain number of iterations. Because these techniques couldn't produce new information and could only remix existing information—the inputs the model sees are still heavily inter-correlated. So we interrupted the training process when the validation loss was no longer improving. Using "segmented dataset", the accuracy and the precision of the model on the "test set" is slightly higher than using "enhanced dataset", and the recall and the AUC is apparently higher. However, it had a lower specificity. The possible reason was that the information of leading the model to hypertension decision mainly came from retinal microvasculature, while the information about non-hypertension also came from other parts of the images other than the blood vessels. Images from "segmented dataset" that contained only retinal blood vessels more likely to be identified as high blood pressure, therefore the recall is higher and the specificity is lower.

To better understand the effect of hypertension on phenotypic traits of retinal microvasculature, we used the images that contained only retinal blood vessels to produce heat maps. As

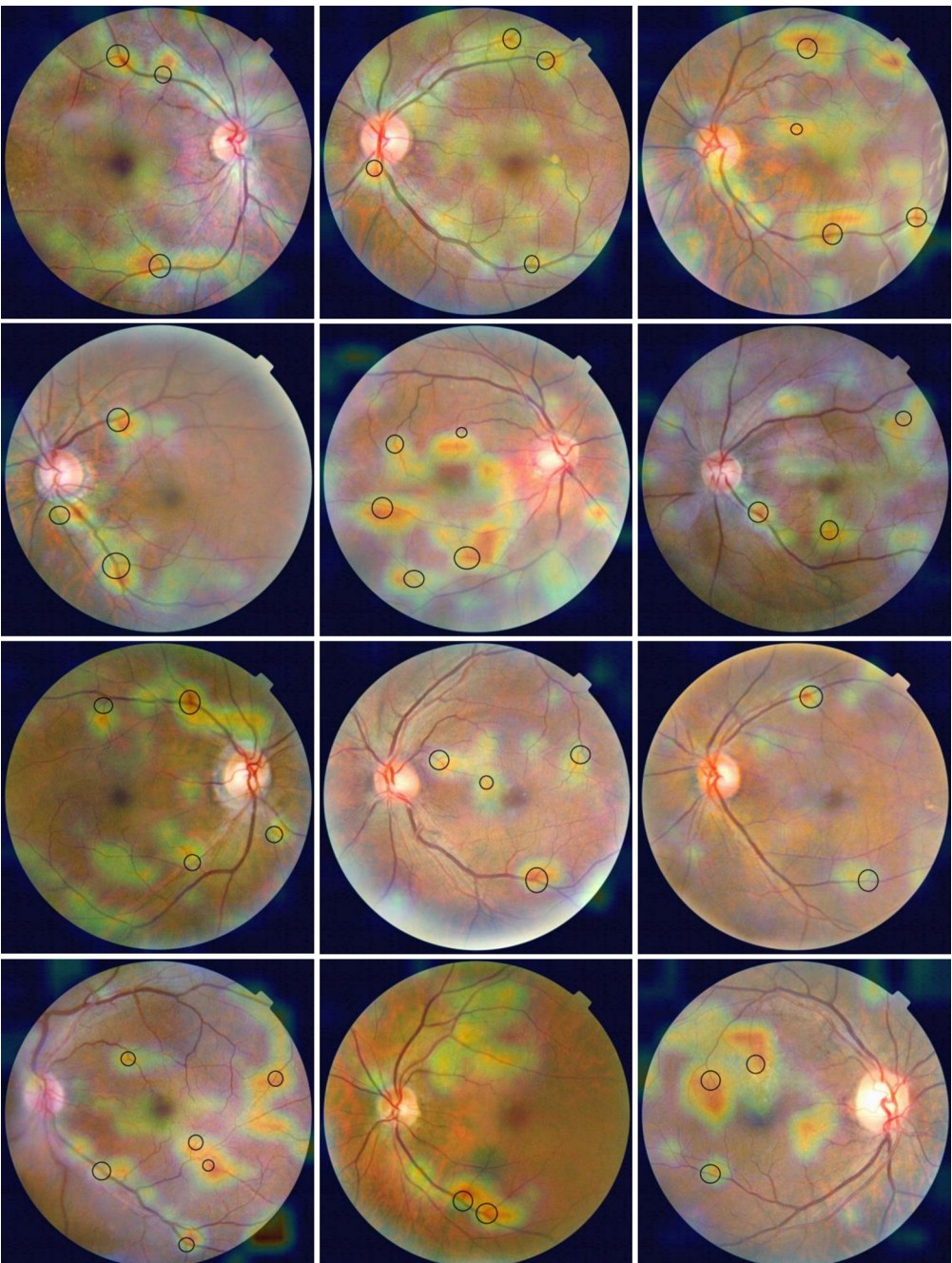

**Fig 4. Heat maps for the class "hypertension".** Arterial/venous bifurcations were within the black ring.

shown in Fig 4, red patchy areas in the heat maps for the class "hypertension" were mainly distributed on or around arterial/venous bifurcations. This indicated that the model identified these regions as being the most important for predicting hypertension. The change of the branching pattern of retinal vessels was probably the most significant among a series of retinal vascular geometric characteristics changes (e.g., caliber, tortuosity, fractal dimension, branching angle) in response to elevated blood pressure. It has been assumed that the branching geometry of blood vessels is governed by definite rules and principles, including principles of

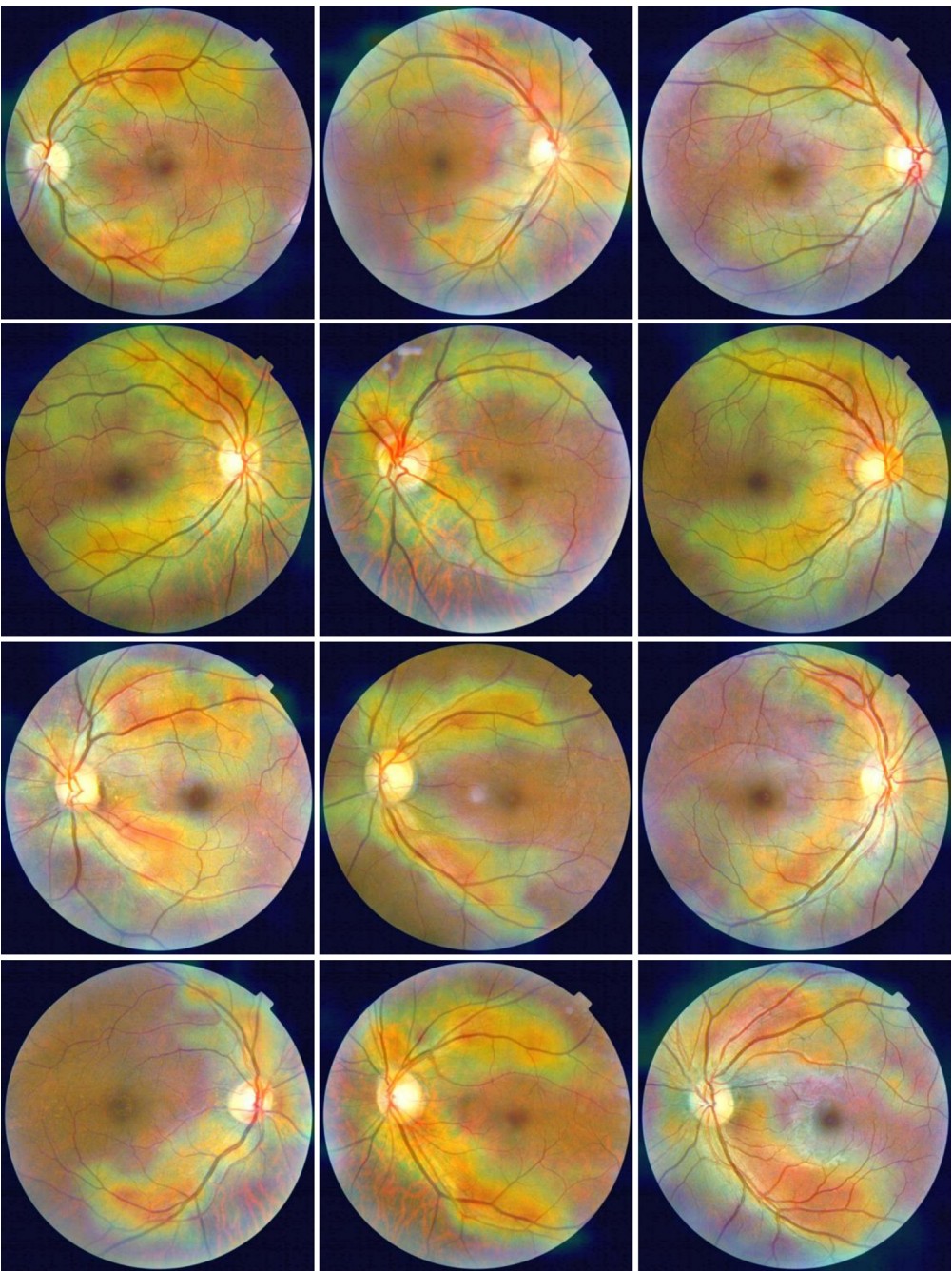

**Fig 5. Heat maps for the class "non-hypertension".**

minimum surface, minimum volume, minimum power, and minimum drag, which related to the physiological function of the cardiovascular system [53]. Based on these optimality principles, measurements of branching angles and the relative diameters of the vessels in arterial bifurcation mainly distributed in the optimum regions. A similar research that measured arterial bifurcations in the retina of a normal human eye and derived quantitative information supported these theoretical studies above [54]. Under different pathological conditions, such as diabetes [55, 56], branching pattern of retinal vessel should change.

One previous study used a semi-automated computer-assisted program to analyze digitized retinal photographs from a total of 1913 hypertensive patients without diabetes, and quantitatively measure the some retinal vascular parameters, including retinal vascular branching angle and retinal vascular branching asymmetry ratio [8]. The results showed that after controlling for other confounding factors, retinal arteriolar branching asymmetry ratios (the ratio of the square of the two branching vessel widths) were independently associated with mean arterial blood pressure, while arteriolar/venular branching angle and venular branching asymmetry ratio were not related to blood pressure. However, in our heat maps, the red areas that had the strongest association with hypertension covered not only the bifurcation of the arteries, but also the veins. That's probably because the segmented images lost color information, and thus unable to classify arteries and veins. For the same reason, some vessel crossings were found to be activated in heat maps. In future work, we will use images that contain only retinal arteriole or venule to construct CNN models to classify hypertension.

Recently, Poplin's research group analyzed datasets from the UK Biobank and EyePACS cohorts using deep learning methods [46]. They trained neural networks to predict known cardiovascular risk factors such as smoking status, systolic blood pressure (SBP). The results showed that the predicted SBP increased linearly with actual SBP until approximately 150 mmHg, but leveled off above that value. They used soft attention to identify the anatomical regions that the model might have been using to make its predictions, and blood vessels were highlighted in the models trained to predict SBP. Our study provided further evidence that the effect of hypertension on retinal microvascular morphology mainly occurred at branching of arterioles. The geometry of retinal arterioles at or near arterial branching may clearly be a factor in the incidence of certain arterial lesions at such junctions.

## Conclusions

The results of this study suggested that the change of the branching pattern of retinal vessel was probably the most significant in response to elevated blood pressure.

## Author Contributions

**Conceptualization:** Wei He.

**Data curation:** Guangzheng Dai, Shasha Liu, Chenguang Zhang.

**Formal analysis:** Guangzheng Dai.

**Investigation:** Guangzheng Dai, Ling Xu, Tiezhu Lin.

**Methodology:** Guangzheng Dai.

**Project administration:** Wei He, Ling Xu.

**Supervision:** Wei He, Ling Xu, Eric E. Pazo, Tiezhu Lin.

**Validation:** Eric E. Pazo, Tiezhu Lin.

**Writing – original draft:** Guangzheng Dai.

**Writing – review & editing:** Eric E. Pazo.

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
