## [Decision Letter · Decision Letter 0]

16 Jan 2020

PONE-D-19-32450

Exploring the effect of hypertension on retinal microvasculature using deep learning on East Asian population: A pilot study employing small database

PLOS ONE

Dear Prof He,

Thank you for submitting your manuscript to PLOS ONE. After careful consideration, we feel that it has merit but does not fully meet PLOS ONE’s publication criteria as it currently stands. Therefore, we invite you to submit a revised version of the manuscript that addresses the points raised during the review process.

SPECIFIC ACADEMIC EDITOR COMMENTS: An expert in the field reviewed this manuscript. There was much interest in this study. However, a comment was made about the small sample size. A goal of PLOS ONE is to publish studies of rigorous experimental design, not necessarily "pilot studies" that may not be reproducible. An attempt should be made to increase the sample size to make this less of a pilot study.

We would appreciate receiving your revised manuscript by Mar 01 2020 11:59PM. To enhance the reproducibility of your results, we recommend that if applicable you deposit your laboratory protocols in protocols.io, where a protocol can be assigned its own identifier (DOI) such that it can be cited independently in the future. For instructions see: http://journals.plos.org/plosone/s/submission-guidelines#loc-laboratory-protocols

We look forward to receiving your revised manuscript.

Kind regards,

Frank T. Spradley

Academic Editor

PLOS ONE

2. Please amend either the abstract on the online submission form (via Edit Submission) or the abstract in the manuscript so that they are identical.

Reviewers' comments:

Reviewer's Responses to Questions

**Comments to the Author**

1. Is the manuscript technically sound, and do the data support the conclusions?

Reviewer #1: Yes

2. Has the statistical analysis been performed appropriately and rigorously? 

Reviewer #1: Yes

3. Have the authors made all data underlying the findings in their manuscript fully available?

Reviewer #1: Yes

4. Is the manuscript presented in an intelligible fashion and written in standard English?

Reviewer #1: Yes

5. Review Comments to the Author

Reviewer #1: Study title: Exploring the effect of hypertension on retinal microvasculature using deep learning on East Asian population: A pilot study employing small database

The study looks good attempt as a pilot study. But it has limitations of low sample size and low sensitivity and specificity.

I have no further comments.

6. PLOS authors have the option to publish the peer review history of their article (what does this mean?). If published, this will include your full peer review and any attached files.

Reviewer #1: Yes: Raba Thapa MD, PhD

---

## [Author Response · Author response to Decision Letter 0]

8 Feb 2020

For the problem about the small sample size, we collected another 407 retinal photographs from patients with a diagnosis of hypertension and 405 retinal photographs from normotensive patients. There were 2012 retinal images including 1007 hypertension images and 1005 non-hypertension images in our study. The dataset size, however, was small for deep learning. So we use some strategies, such as reducing the size of the model, using dropout layers and data augmentation, to mitigate overfitting and maximize generalization. After increasing sample size, for “segmented dataset” the evaluation parameters (accuracy, specificity, precision, recall, the area under the receiver operating characteristic curve) changed little, while for “enhanced dataset” the change was comparatively large, as shown in the following table. The possible reason was that images from “segmented dataset” contained only retinal microvasculature and the information of leading the model to hypertension decision mainly came from retinal microvasculature. But images from “enhanced dataset” contained a lot of additional information besides retinal microvasculature and the model learns representations that are not specific to hypertension. Accordingly we inferred that with increasing sample size, for “segmented dataset” the result will not be much different.

---

## [Decision Letter · Decision Letter 1]

24 Feb 2020

Exploring the effect of hypertension on retinal microvasculature using deep learning on East Asian population

PONE-D-19-32450R1

Dear Dr. He,

We are pleased to inform you that your manuscript has been judged scientifically suitable for publication and will be formally accepted for publication once it complies with all outstanding technical requirements.

With kind regards,

Frank T. Spradley

Academic Editor

PLOS ONE

Reviewers' comments:

Reviewer's Responses to Questions

**Comments to the Author**

1. If the authors have adequately addressed your comments raised in a previous round of review and you feel that this manuscript is now acceptable for publication, you may indicate that here to bypass the “Comments to the Author” section, enter your conflict of interest statement in the “Confidential to Editor” section, and submit your "Accept" recommendation.

Reviewer #1: All comments have been addressed

2. Is the manuscript technically sound, and do the data support the conclusions?

Reviewer #1: Yes

3. Has the statistical analysis been performed appropriately and rigorously? 

Reviewer #1: Yes

4. Have the authors made all data underlying the findings in their manuscript fully available?

Reviewer #1: Yes

5. Is the manuscript presented in an intelligible fashion and written in standard English?

Reviewer #1: Yes

6. Review Comments to the Author

Reviewer #1: I congratulate the authors for good attempt. The addition of more pictures have made the outcome better.

I have no comments for this manuscript.

7. PLOS authors have the option to publish the peer review history of their article (what does this mean?). If published, this will include your full peer review and any attached files.

Reviewer #1: No

---

## [Editor Report · Acceptance letter]

27 Feb 2020

PONE-D-19-32450R1 

Exploring the effect of hypertension on retinal microvasculature using deep learning on East Asian population 

Dear Dr. He:

I am pleased to inform you that your manuscript has been deemed suitable for publication in PLOS ONE. Congratulations! Your manuscript is now with our production department. 

With kind regards,

on behalf of

Dr. Frank T. Spradley 

Academic Editor

PLOS ONE